# Combined Proteomic and Physiological Analysis of Chloroplasts Reveals Drought and Recovery Response Mechanisms in *Nicotiana benthamiana*

**DOI:** 10.3390/plants10061127

**Published:** 2021-06-02

**Authors:** Silin Chen, Ping Li, Shunling Tan, Xiaojun Pu, Ying Zhou, Keming Hu, Wei Huang, Li Liu

**Affiliations:** 1Hubei Collaborative Innovation Center for Green Transformation of Bio-Resources, State Key Laboratory of Biocatalysis and Enzyme Engineering, Hubei Key Laboratory of Industrial Biotechnology, School of Life Sciences, Hubei University, Wuhan 430062, China; chensilin@mail.kib.ac.cn; 2Yunnan Key Laboratory for Wild Plant Resources, Department of Economic Plants and Biotechnology, Kunming Institute of Botany, Chinese Academy of Sciences, Kunming 650204, China; tanshunling@mail.kib.ac.cn (S.T.); puxiaojun@mail.kib.ac.cn (X.P.); 3Key Laboratory for Forest Resources Conservation and Utilization in the Southwest Mountains of China Ministry of Education, Southwest Forestry University, Kunming 650224, China; liping@mail.kib.ac.cn; 4College of Life Science and Health, Wuhan University of Science and Technology, Wuhan 430065, China; zhouying0613@wust.edu.cn; 5Jiangsu Key Laboratory of Crop Genomics and Molecular Breeding, Key Laboratory of Plant Functional Genomics of the Ministry of Education, College of Agriculture, Yangzhou University, Yangzhou 225009, China; hukm@yzu.edu.cn; 6Co-Innovation Center for Modern Production Technology of Grain Crops of Jiangsu Province, Key Laboratory of Crop Genetics and Physiology of Jiangsu Province, Yangzhou University, Yangzhou 225009, China

**Keywords:** *Nicotiana benthamiana*, drought stress, recovery, chloroplast, proteomics, physiological analyses

## Abstract

Chloroplasts play essential roles in plant metabolic processes and stress responses by functioning as environmental sensors. Understanding chloroplast responses to drought stress and subsequent recovery will help the ability to improve stress tolerance in plants. Here, a combined proteomic and physiological approach was used to investigate the response mechanisms of *Nicotiana benthamiana* chloroplasts to drought stress and subsequent recovery. Early in the stress response, changes in stomatal movement were accompanied by immediate changes in protein synthesis to sustain the photosynthetic process. Thereafter, increasing drought stress seriously affected photosynthetic efficiency and led to altered expression of photosynthesis- and carbon-fixation-related proteins to protect the plants against photo-oxidative damage. Additional repair mechanisms were activated at the early stage of recovery to restore physiological functions and repair drought-induced damages, even while the negative effects of drought stress were still ongoing. Prolonging the re-watering period led to the gradual recovery of photosynthesis at both physiological and protein levels, indicating that a long repair process is required to restore plant function. Our findings provide a precise view of drought and recovery response mechanisms in *N. benthamiana* and serve as a reference for further investigation into the physiological and molecular mechanisms underlying plant drought tolerance.

## 1. Introduction

Chloroplasts are one of the most important semi-autonomous organelles in green plants. Beside serving as the sites of photosynthesis, chloroplasts also have important roles as sensors of environmental stimuli and as hosts of numerous other cellular vital processes, including retrograde signaling [1], sulfate assimilation [2], redox adjustment [3], carotenoid biosynthesis [4], and fatty acid biosynthesis [5]. These processes are closely interconnected and play irreplaceable parts in regulating metabolic processes and stress responses in plants [6].

As a major and frequent stress factor, drought has significant effects on the distribution and production of crops and ornamental plants [7]. Drought stress adversely affects plant growth and development by altering their metabolic and physiological processes, especially photosynthesis [8]. Photosynthesis is one of the most crucial metabolic processes in plants; thus, sustaining its efficiency is essential for plant survival under changing environmental conditions. As reported in previous studies, drought can cause an imbalance between the absorption and utilization of light [9]. This drought-induced imbalance leads to severe over-accumulation of reactive oxygen species, which in turn can damage the plant photosynthetic apparatus, creating a vicious cycle [10]. Over the long course of evolution, plants have evolved sophisticated mechanisms to respond to drought stress and repair drought-induced damage [11]. However, the repair and rearrangement processes require time; therefore, the subsequent recovery is also essential for plants to successfully live through the drought stress.

In recent years, proteomic analysis has proven to be an effective strategy for investigating the molecular mechanisms involved in plant drought responses because it provides a direct assessment of the molecules involved in crucial cellular and physiological functions [12]. Although this topic has been intensively investigated in many plant species [13,14,15], few studies have focused on the process of recovery from drought, and especially the dynamic changes in the chloroplast proteome that this entails.

Drought tolerance, a complex trait that involves sophisticated regulatory networks, is required for plant survival under rapidly changing climate conditions. Understanding the response of the chloroplast to drought stress and re-watering will facilitate the improvement of drought tolerance in plants. In the current study, we investigated the mechanisms underlying the responses of chloroplasts to drought stress and subsequent recovery. Specifically, we combined physiological and isobaric tags for relative and absolute quantitation (iTRAQ)-based proteomic approaches to investigate this process in the model plant *N. benthamiana*. Our findings lay the foundation for further investigation of the physiological and molecular mechanisms underlying drought tolerance in plants.

## 2. Results

### 2.1. Drought Stress in Nicotiana Benthamiana Is Reflected in Dynamic Changes in the Chloroplast Proteome

To investigate the dynamic changes that occur in the chloroplast proteome in response to drought stress and subsequent recovery, we performed iTRAQ-based chloroplast proteome analysis in *N. benthamiana* on drought and re-watering treatment samples from five experimental groups (Figure 1). We detected 1087 chloroplast proteins in total. Proteins showing a difference in abundance ratio of >1.5 were considered to be upregulated, whereas proteins with a difference in abundance ratio of <0.667 were considered to be downregulated under different conditions. Based on this standard, 329 differentially expressed proteins (DEPs) were identified comparing D0 vs. D1, D0 vs. D5, D0 vs. R1, and D0 vs. R3 (Appendix A).

To investigate the functions of these DEPs, we performed a Kyoto Encyclopedia of Genes and Genomes (KEGG) pathway enrichment analysis (Figure 2). A large proportion of DEPs (36.2%) were markedly enriched in metabolic pathways. Proteins involved in processes of photosynthesis, photosynthesis-antenna proteins, glyoxylate and dicarboxylate metabolism, carbon fixation in photosynthetic organisms, and porphyrin and chlorophyll metabolism, accounted for 36.6% of the total DEPs.

We grouped the 329 DEPs into eight categories (expression profiles) based on their differential accumulation patterns (Figure 3). Proteins in profiles 1 and 7 significantly increased in abundance with increasing degrees of water deficit, reaching their peaks during the early phase of recovery (R1). Of these, 30 proteins are involved in photosynthesis and 26 are photosynthesis-antenna proteins. By contrast, proteins in profiles 2, 4, and 5 showed a downward trend in abundance during the course of drought treatment (D1 and D5), and most reached their lowest levels at R1. Proteins in profiles 3, 6, and 8 were subject to more complex dynamic changes. The expression of proteins in profile 6 was induced by light drought-stress treatment (D1) but decreased significantly with increasing drought stress (D5). The accumulation of proteins in profile 3 was significantly repressed during both D1 and R1. Finally, most proteins in profile 8 increased in abundance during D5 and R3 and were not significantly affected by a slight water deficit. These results provide a global view of drought and recovery response mechanisms of *N. benthamiana* chloroplasts at the proteome level.

### 2.2. Changes in Protein Synthesis Are Triggered at the Early Stage of Drought Stress

We generated Venn diagrams showing the number, percentage, and overlap of chloroplast proteins that differentially accumulated during drought stress and re-watering (Figure 4A). Among these proteins, 13 showed significant decreases during this treatment. In contrast, 13 proteins underwent dramatic increases in abundance during the early stage of drought stress (D1). Five of these proteins participate in photosynthesis, including chloroplast ferredoxin I (Fd I) and the ribulose-1,5-bisphosphate carboxylase/oxygenase (Rubisco) small subunits (RbcS). As expected, physiological responses to water deficit lagged behind changes in protein levels (Figure 5A,B). Stomatal conductance (*g*_s_) decreased slightly (3.5%) in response to mild drought stress (D1), whereas the net photosynthesis rate (*A*_N_) and the rate of linear electron transport (ETR) were unchanged during this period. Thus, modulation of protein synthesis is the primary type of change triggered during the early stage of drought stress.

### 2.3. Protective Mechanisms Are Comprehensively Activated in Response to Increasing Drought Stress

Compared with D0, at D5, when drought stress was more severe, the expression of 81 proteins was induced and that of 99 proteins was repressed. Most of the upregulated proteins were grouped into profiles 1, 7, and 8. The number of upregulated proteins specifically involved in photosynthesis (including the antenna proteins of photosynthesis) increased with the increasing drought stress, ranging from 4 proteins at D1 to 32 at D5 (Figure 4B). Numerous key enzymes in carotenoid biosynthesis, porphyrin and chlorophyll metabolism, and carbon fixation were considerably downregulated in plants subjected to severe water deficit conditions, including ζ-carotene desaturase (ZDS), Rubisco large subunit (RbcL), NADPH-protochlorophyllide oxidoreductase (POR1), and NAD-malate dehydrogenase. As compared to mild drought stress, five days of drought treatment resulted in more extensive changes in physiological parameters: at D5, the *A*_N_ and ETR decreased continuously, reaching values of only 51.1% and 79.1% those at D0, respectively (Figure 5A, B). In addition, increasing drought stress resulted in a dramatic decrease in *g*_s_, from 80.9% after 3 days to 24.3% after 5 days of water deficit. These results suggested that lengthier drought stress led to notable physiological changes in the plants.

### 2.4. Negative Effects of Drought Stress Are Still Ongoing at the Early Stage of Re-watering

During the early recovery phase (R1), the levels of 135 proteins, mostly in profiles 1 and 7, were significantly higher than the basal levels in D0 (Figure 4B). Among these proteins, 57 were related to the process of photosynthesis (including 26 photosynthesis-antenna proteins). Of these photosynthesis-related proteins, 24 showed clear induction only at R1, including several key subunits of the photosystem I and photosystem II reaction centers: PsbB, PsbC, PsbE, and PsaE (A and B). In contrast, the levels of 149 proteins decreased dramatically under this treatment, particularly proteins that function in carbon fixation in photosynthetic organisms and in the metabolic processes of porphyrin and chlorophyll. Notably, among these downregulated proteins, significant decreases in the levels of plastid transketolase, NADP-dependent malate dehydrogenase, and NAD-dependent glyceraldehyde-3-phosphate dehydrogenase, were observed only at R1. Marked increases in the photosynthetic parameters *A*_N_, ETR, and *g*_s_ were observed in the plants after re-watering for 8 h. During this period, the *A*_N_ and ETR increased by 31.5% and 8.5%, respectively, and the *g*_s_ increased by 86.5% compared to the values at D5. These results indicated that the negative effects of drought stress were still ongoing, and at the same time further repair mechanisms were activated to restore the physiological functions and repair the drought-induced damage.

### 2.5. Recovery Is a Long Process

At the late stage of recovery, the levels of 54 proteins, including 13 photosynthesis-related proteins, were still higher than the background (D0) levels (Figure 4B). Nevertheless, compared to R1, all these photosynthesis-related proteins were less abundant, to different degrees, during late recovery. In contrast, the level of some key proteins that participate in porphyrin and chlorophyll metabolism, such as NADPH: protochlorophyllide oxidoreductase, the magnesium-chelatase subunit chlI, and Mg protoporphyrin IX chelatase, were higher at R3 than at D0. Surprisingly, although the ETR decreased by approximately 21% at D5, this value recovered completely to its original level at R3’ (Figure 5B). However, after re-watering for 56 h, the *g*_s_ and *A*_N_ only recovered to 64% and 86.5% of their original levels, respectively (Figure 5A). During the re-watering period, the gradual recovery of photosynthesis at both the physiological and protein levels highlighted that repair is a long process for plants.

## 3. Discussion

Drought stress can provoke changes in many biological processes and many aspects of cellular metabolism, especially protein synthesis [16]. In the current study, we investigated 26 proteins whose expression levels were significantly affected by mild drought stress. Among these proteins, 5 photosynthetic proteins were upregulated at D1, especially the small subunit of Rubisco (RbcS). Rubisco, a key photosynthetic enzyme, is directly linked to photosynthetic metabolism and plays a crucial role in maintaining carbon–nitrogen balance in plants [17]. The necessity of RbcS for maximal catalytic efficiency of the enzyme has been verified in previous studies [18]. Compared with the large subunits of Rubisco (RbcL), RbcS shows a higher binding affinity for CO_2_ and therefore can serve as a CO_2_ reservoir [19]. Another noteworthy protein for which accumulation was significantly induced during early drought stress was chloroplast ferredoxin I (Fd I). Fd I is involved in numerous essential metabolic processes and plays an indispensable role in electron transfer in chloroplasts [20]. At the physiological level, only the stomatal conductance (*g*_s_) showed a slight decrease (3.5%) at D1. As the stages of water deficit progressed (from D1 to D3), the net photosynthesis rate (*A*_N_) and the linear electron transport rate (ETR) also gradually decreased. These results suggest that *N. benthamiana* is sensitive to water deficit: even if no obvious physiological changes were observed, immediate responses were provoked in many biological processes, especially in protein synthesis, to sustain the process of photosynthesis under mild drought stress.

With the increasing degree of drought, *A*_N_ and ETR were seriously affected, as expected. Surprisingly, *g*_s_ drastically decreased by D3 (to 80.9% of the value at D0) and reached its lowest level at the late stage of drought stress (D5) (24.3% of the value at D0). The levels of both RbcL and RbcS decreased significantly under severe drought stress (at D5), potentially leading to a further decrease in Rubisco activity [21]. Furthermore, we detected a positive correlation between the decreasing levels of the Rubisco subunits and the decline of *A*_N_, suggesting that limited carboxylase capacity might be a primary cause for the inhibition of photosynthesis efficiency that occurs under severe drought stress. The decreased abundance of Rubisco accompanied by marked decreases of other photosynthesis-related proteins, including ATP synthase CF1 alpha subunit, fructose-1,6-bisphosphatase, and NAD-malate dehydrogenase, have similarly been observed in previous research [22]. In addition to causing prominent changes in photosynthetic metabolism, drought stress can lead to the degradation of photosynthetic pigments and inhibit chlorophyll synthesis [23], which is consistent with our results. In the current study, seven proteins with essential roles in porphyrin and chlorophyll metabolism were significantly repressed by severe drought stress. At D5, we also detected the repressed expression of ζ-carotene desaturase (ZDS), an essential enzyme in the carotenoid biosynthesis pathway that plays crucial roles in plant growth and development. The inhibition of ZDS activity results in a significant decrease in chlorophyll content and an accumulation of superoxide, which in turn cause photo-oxidative damage to the photosynthetic apparatus [24]. Under these conditions, protective mechanisms are activated in plants to maintain photosynthesis and confer resistance to drought-induced damage, ultimately enhancing plant survival. PsbS, a small subunit of photosystem II, interacts with light-harvesting complex II (LHC-II) and protect plants from photodamage by efficiently dissipating excessive light energy via non-photochemical quenching (NPQ) [25]. The significant up-regulation of PsbS in *N. benthamiana* at D5 suggested that NPQ was enhanced under severe drought stress. Similar results have been reported in *Arabidopsis thaliana* [26]. Our results suggest that drought stress directly inhibits photosynthetic electron transfer in plants and leads to a significant decrease in ETR. Indeed, electron transport capacity can affect the regeneration of ribulose-1,5-bisphosphate (RuBP), the substrate of Rubisco [27]. In response, the cytochrome b6-f complex iron-sulfur subunit 2 (PetC), plastocyanin (PetE), and the ATP synthase CF1 alpha subunit accumulate under drought stress to sustain light-induced electron transfer, as detected after 5 days of water-deficit treatment [28]. This process also protects plants from photo-oxidative damage by regulating luminal acidification and enhancing the thermal dissipation capacity [29,30]. Together these proteins constitute a sophisticated network that is essential in plant responses to drought stress.

Recovery after water deficit is a dynamic, complex process that is necessary for the survival of plants under drought stress. Our physiological analysis indicated that during the early stage of recovery, the photosynthetic process partially but rapidly recovered after drought stress was removed. However, proteomic analysis revealed that the chloroplast proteome continued to be modified during this period. Compared with that at D0, the expression of most photosynthesis-related proteins (accounting for approximately 93.4% of the total) increased after re-watering for 8 h. Among these proteins, 24 were only obviously induced at R1, including key subunits of the photosystem I and II reaction centers (PsbB, PsbC, PsbE, and PsaE A and B). In contrast, the number of proteins whose expression was downregulated increased from 99 at D5 to 149 at R1. The accumulation of DEPs that play crucial roles in porphyrin and chlorophyll metabolism was significantly repressed during the early phase of recovery. Moreover, 43 proteins, including NADP-dependent-malate dehydrogenase, NAD-dependent glyceraldehyde 3-P dehydrogenase, catalase, plastid transketolase, and glutamine synthetase, were repressed only at R1. The finding that glutamine synthetase, a key enzyme that regulates the nitrogen cycle in chloroplasts by assimilating ammonium produced by photorespiration into glutamine [31], was downregulated at R1 suggests that the enhanced photosynthetic efficiency at this stage could be attributed to increased CO_2_ concentrations. This notion is consistent with the results of physiological analysis, and a similar effect of drought stress was reported previously [32]. However, the decline in NADP–malate dehydrogenase levels combined with the obvious decrease in catalase levels would be expected to lead to a significant increase in H_2_O_2_ levels [33]. These results indicate that even if appreciable recovery was detected at the physiological level, the inhibition of metabolic processes induced by drought stress were still exacerbated during the early stage of recovery, suggesting that other repair mechanisms covering a wider range of photosynthetic processes were also activated at the same time.

With the prolonging of re-watering, the photosynthesis process recovered efficiently. At this time, the ETR had totally recovered to its original level; nevertheless, the *A*_N_ only recovered to 86.5% of its original value. The incomplete resumption of photosynthetic efficiency at R3 might be attributed to the restriction of stomatal conductance. An analysis of the chloroplast proteome at R3 revealed that the modification and adjustment of multiple metabolic processes to repair drought-induced damage was still in progress at that time. The abundance of most photosynthesis-related proteins, such as Fd I, PsbE, and the ATP synthase CF1 alpha subunit, declined gradually and tended to be stable at the late stage of re-watering. In contrast, porphyrin and chlorophyll metabolism were activated to balance the biosynthesis and degradation of photosynthetic pigments [32]. These results suggest that a wide range of metabolic processes recovered gradually after drought stress was relieved, requiring more time for rebuilding and repair in plants.

## 4. Materials and Methods

### 4.1. Plant Materials and Drought Treatment

*Nicotiana benthamiana* plants were cultivated in greenhouse set at 30 ℃/20 ℃ with light intensity of 180 μmol m^−2^s^−1^ and 14 h/10 h photoperiod (light/dark). Forty-day-old plants were used for drought treatment. Drought stress was induced by withholding water until the relative water contents decreased to approximately 30% (D0–D8). D0 was the day before drought treatment, and D1–D8 stood for the day of drought. After drought treatment, plants were re-watered, and two samplings (R1: re-watered for 8 h; R3: re-watered for 48 h) were harvested for further proteomic analysis.

For the analysis of photosynthesis characteristics, *N. benthamiana* grown in a greenhouse under the above conditions were exposed to drought for 5 days (D0–D5) until the relative water contents reached the level of the above conditions. After drought treatment, plants were re-watered, and three samplings were harvested in light cycles (R1: re-watered for 8 h; R2’: re-watered for 32 h; R3’: re-watered for 56 h).

### 4.2. Chloroplast Isolation and Protein Extraction

Chloroplasts of different treatment materials were isolated according to the method of Li et al. [34] Then the isolated chloroplasts were resuspended and sonicated in extraction buffer (4% *w*/*v* SDS, 100 mM Tris-HCl, 1 mM DTT, pH 7.6), followed by extraction according to Kosmala’s method [35].

### 4.3. ITRAQ Labeling and Liquid Chromatography-Tandem Mass Spectrometry (LC-MS/MS) Analysis

Protein samples were labeled with an 8-plex iTRAQ kit (AB SCIEX, Foster City, CA, USA) following the manufacturer’s instructions. Q Exactive™ hybrid quadrupole-Orbitrap (Thermo Fisher Scientific, San Jose, CA, USA) with a Higher Energy Collisions Dissociation (HCD) model were then used to analyze and quantify the dynamic changes of the proteome. To ensure adequate coverage, three biological replicates were collected. The raw data were processed using MASCOT (v.2.2) and searched against the *Nicotiana benthamiana* protein database (NCBI) with a false discovery rate (FDR) <1%.

### 4.4. Bioinformatics Analysis

KOBAS 3.0 was used to search the Kyoto Encyclopedia of Genes and Genomes (KEGG) database and annotate the protein pathway (*Nicotiana tabacum*) [36]. The expression patterns of differentially expressed proteins (DEPs) during the treatments were characterized by TCseq [37]. The Venn diagrams of differentially expressed proteins were constructed by Venny 2.1.0 [38].

### 4.5. Analysis of Photosynthesis Characteristics

A Li-6400 Portable Photosynthesis System (Li-Cor, Lincoln, NE, USA) was used to monitor the photosynthesis characteristics during the drought and re-watering treatments. The net photosynthesis rate (*A*_N_), stomatal conductance (*g*_s_), and the rate of linear electron transport (ETR) was recorded at the photon flux density of 500 μmol m^−2^ s^−1^ at 28 °C. Five independent biological replicates with three times repeated were applied to each experiment. Statistical analysis was performed using SPSS 22.0 (IBM, Armonk, NY, USA). All data are represented as the means ± SE and *p*-value < 0.05 was considered significant.

## 5. Conclusions

Our findings suggest that the modulation of stomatal movement combined with the fine-tuning of the chloroplast proteome represents a useful strategy to sustain metabolic processes in plants in response to drought stress. We also demonstrated that recovery after water deficit is a dynamic and sophisticated process. Our findings provide a precise view of the mechanisms underlying the response to drought stress and subsequent recovery in *N. benthamiana* chloroplasts. These data will facilitate the functional dissection of the molecular mechanism underlying chloroplast responses to environmental stress in *N. benthamiana*, paving the way for improving drought tolerance in plants.

## Figures and Tables

**Figure 1 plants-10-01127-f001:**
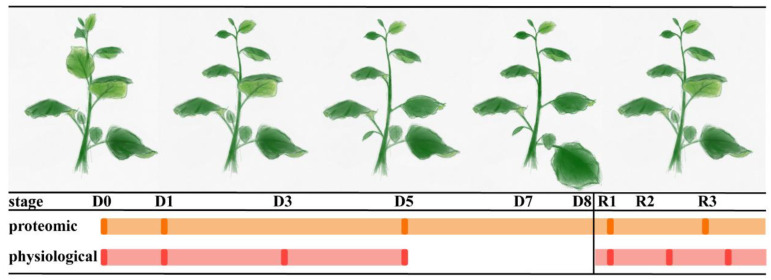
An illustration of *Nicotiana benthamiana* treatment stages and sampling points. D0: no-drought stress, D1–D8: 1 d–8 d of drought treatment, R1–R3: 8 h–48 h of re-watering. Bars were represented as the duration of treatments. Samples obtained at the shown points were used for proteomic and physiological analysis, respectively.

**Figure 2 plants-10-01127-f002:**
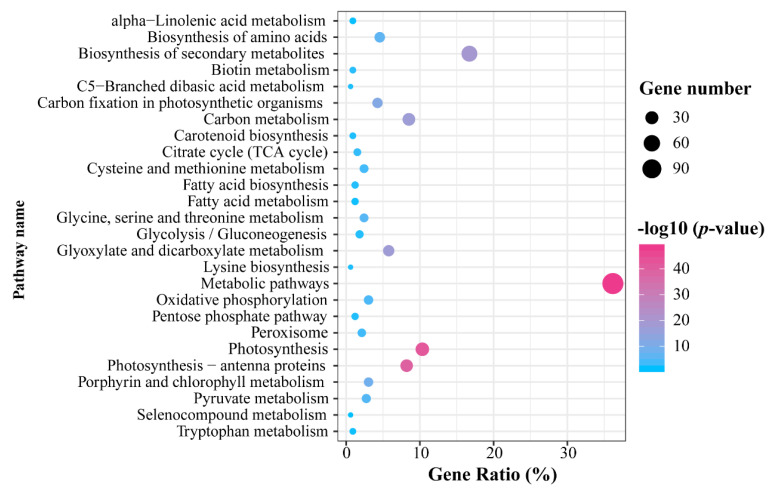
KEGG pathway enrichment analysis of total differentially expressed proteins.

**Figure 3 plants-10-01127-f003:**
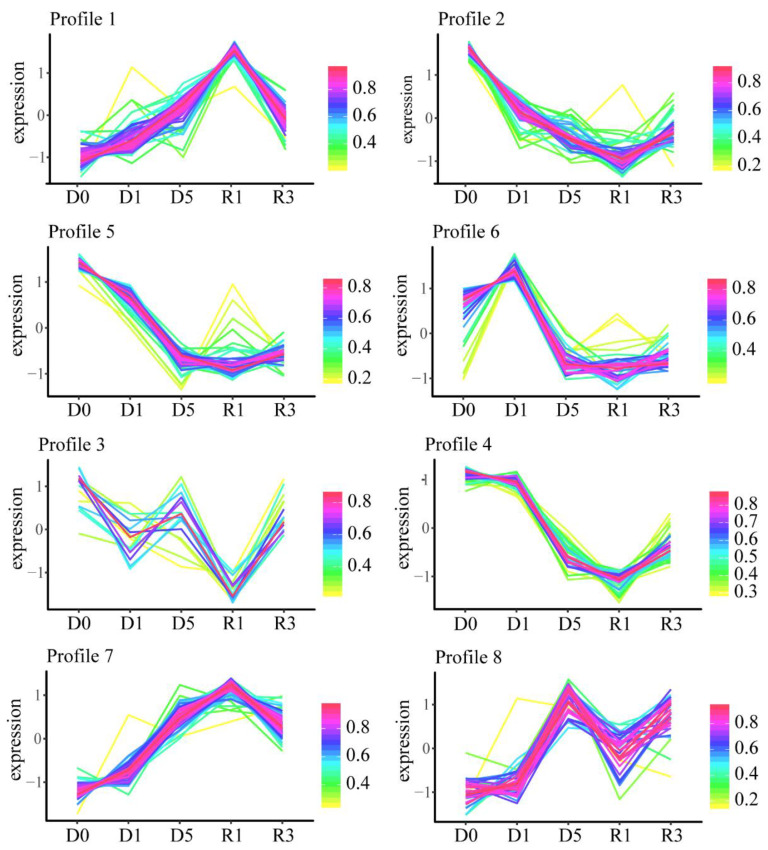
Differential accumulation patterns of total differentially expressed proteins.

**Figure 4 plants-10-01127-f004:**
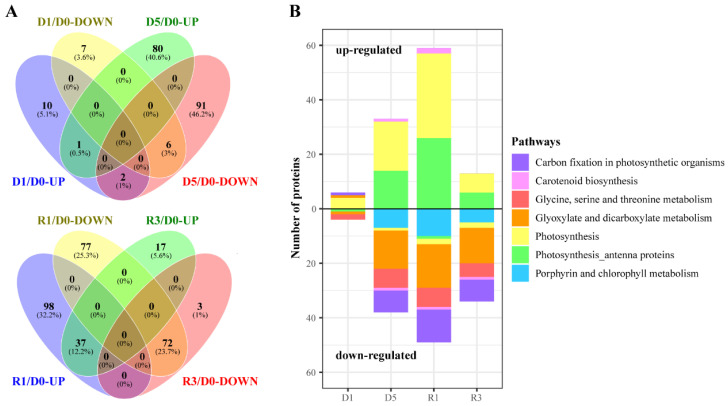
Analysis of differentially expressed proteins at each stage. (**A**) Venn diagram. (**B**) Pathway enrichment analysis.

**Figure 5 plants-10-01127-f005:**
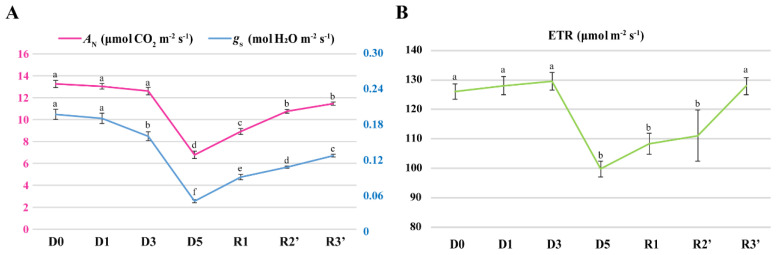
Photosynthesis characteristics analyses. (**A**) Analysis of net photosynthesis rate (*A*_N_) and stomatal conductance (*g*_s_). (**B**) Analysis of rate of linear electron transport (ETR). D0: no-drought stress, D1–D5: 1 d–5d of drought treatment, R1: re-watered for 8 h; R2’: re-watered for 32 h; R3’: re-watered for 56 h. Values are expressed as means ± SE, n = 5. Statistically different values (*p* < 0.05) are indicated by different letters.

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
