# Peer review of "Combined Proteomic and Physiological Analysis of Chloroplasts Reveals Drought and Recovery Response Mechanisms in Nicotiana benthamiana"

_plants, 2021, doi:10.3390/plants10061127_

Round 1
Reviewer 1 Report
The manuscript by Chen and co-authors investigates the proteome of chloroplasts during a period of drought stress and recovery and integrates this information with physiological parameters. The authors have designed an adequate experiment, executed successfully and analyzed the collected data. Drought stress has been extensively studied in many plant species; however, this research provides valuable information about the metabolic processes occasioned in the chloroplast during and after the stress and integrates this information with the photosynthetic process. Overall, I believe this manuscript is important to deepen in the responses of N. benthamiana to drought stress, specifically at the chloroplast, and provides some valuable information to decipher plants mechanism to tolerate drought stress. However, I have some minor concerns:
- The authors presented the results about physiological traits (An, gs and ETR) through the different results subsections to link the information with the results of the proteomic analysis. However, I would prefer the physiological results grouped in a subsection.
- In line 246: “Our physiological analysis indicated that during the early stage of recovery, the photosynthetic process recovered rapidly after drought stress was removed.” Although it is showed a recovery of An, the values are still lower than at the beginning and not even in R3 An reach the initial values. Although, the authors further discuss this issue and suggest that this is caused by the low stomatal aperture, this is not fully demonstrated. That sentence should be modified and clarify that photosynthetic process it is not rapidly fully recovered.
- In line 266-267, authors made a conclusion based on the increase in oxidative stress: “These results indicate that even if appreciable recovery was detected at the physiological level, the inhibition of metabolic processes and increase in oxidative stress induced by drought stress…”. However, there is not direct or indirect measurements of oxidative stress or ROS levels. Although they previously suggest that, under drought stress conditions, the decrease of Catalase would lead to an increase of H2O2, this is not demonstrated. Please modify that sentence
- In lines 323-325 the authors concluded: “We also demonstrated that recovery after water deficit is a dynamic, sophisticated process which is essential for plant survival following drought stress.”. However, their results do not show information about damage occasioned in plants or about survival rate and do not present any results that directly support this conclusion. That sentence should be modified.
- In Material and methods section, explain or cite Arkadiusz's method (line 294), please.
Reviewer 2 Report
This manuscript deals with the chloroplast responses to drought at the proteomic level, but also with a small physiological contribution. The main and original aspects of this article are the time required for the repair of chloroplasts after drought and the changes suffered during the process. On the other hand, as this manuscript combines physiological and proteomic aspects linked to water stress, it is of interest to both plant physiologists and ecophysiologists, as well as to geneticists and proteomics researchers.
This study is of great interest and the topic is within the scope of the journal. The manuscript is very well structured and written, and the bibliography is very up-to-date. So, I can recommend this manuscript for publication after a minor revision.
In my opinion, the editor will be able to evaluate the revised version with these proposed changes, without the need to resubmit it to a reviewer.
Also, see the attached file. It contains paragraphs highlighted in yellow and comments.
